# Voltammetric pH Measurements Using Azure A-Containing Layer-by-Layer Film Immobilized Electrodes

**DOI:** 10.3390/polym12102328

**Published:** 2020-10-12

**Authors:** Kazuhiro Watanabe, Kyoko Sugiyama, Sachiko Komatsu, Kentaro Yoshida, Tetsuya Ono, Tsutomu Fujimura, Yoshitomo Kashiwagi, Katsuhiko Sato

**Affiliations:** 1Faculty of Pharmaceutical Science, Tohoku Medical and Pharmaceutical University, 4-4-1 Komatsushima, Aoba-ku, Sendai, Miyagi 981-8558, Japan; kyoko.sugiyama@tohoku-mpu.ac.jp (K.S.); sachicom@tohoku-mpu.ac.jp (S.K.); tfujitsu@tohoku-mpu.ac.jp (T.F.); 2School of Pharmaceutical Sciences, Ohu University, 31-1 Misumido, Tomita-machi, Koriyama, Fukushima 963-8611, Japan; k-yoshida@pha.ohu-u.ac.jp (K.Y.); t-ono@pha.ohu-u.ac.jp (T.O.); y-kashiwagi@pha.ohu-u.ac.jp (Y.K.); 3Department of Creative Engineering, National Institute of Technology, Tsuruoka College, 104 Sawada, Inooka, Tsuruoka, Yamagata 997-8511, Japan

**Keywords:** Azure A, pH sensor, layer-by-layer deposited film

## Abstract

pH is one of the most important properties associated with an aqueous solution and various pH measurement techniques are available. In this study, Azure A-modified poly(methacrylic acid) (AA-PMA) was synthesized used to prepare a layer-by-layer deposited film with poly(allylamine hydrochloride) (PAH) on a glassy carbon electrode via electrostatic interactions and the multilayer film-immobilized electrode was used to measure pH. Cyclic voltammetry (CV) and differential pulse voltammetry (DPV) measurement were performed. Consequently, the oxidation potential of AA on the electrode changed with pH. As per Nernst’s equation, because H^+^ ions are involved in the redox reaction, the peak potential shifted depending on the pH of the solution. The peak potential shifts are easier to detect by DPV than CV measurement. Accordingly, using electrochemical responses, the pH was successfully measured in the pH range of 3 to 9, and the electrodes were usable for 50 repeated measurements. Moreover, these electrochemical responses were not affected by interfering substances.

## 1. Introduction

Layer-by-layer deposition is a technique used to form a nanometer-thick multilayer film on a solid substrate surface, such as metals and glass, via electrostatic interactions by alternately immersing the substrate in a polycation and polyanion solutions (Scheme 1) [1]. Formation of a thin multilayer film can be both driven by electrostatic interaction and by other types of interactions, such as biological affinity, e.g., avidin–biotin bonds [2,3,4], sugar–lectin bonds [5]; hydrogen bonds [6,7]; diol–phenylboronic acid bonds [8,9]; guest–host interactions [10]; and other low energy physical bonds [11,12,13]. Thus, a functional thin film can be formed from synthetic polymers and other materials, such as proteins, such as enzymes [14,15], polysaccharides [16,17], supramolecular compounds [18], and nanoparticles [19]. Furthermore, functional molecules can be easily immobilized in a film by modifying them with a polymer chain. For example, it has been reported that a multilayer thin film composed of a phenylboronic acid-modified polymer and polyvinyl alcohol decomposes in response to sugar and hydrogen peroxide [20]. The multilayer thin films composed of functional organic molecules, enzymes, and lectins also respond to substrates, such as pH [21,22], electrical potential [23], glucose, lactate, and hydrogen peroxide [24,25,26]. Such functional multilayer thin films could be useful for separation and purification [27,28], sensors [29,30], and drug delivery systems [31,32,33,34].

In this study, a simple pH sensor was fabricated by immobilizing Azure A (AA) whose oxidation and reduction potentials change depending on the solution pH using a layer-by-layer deposition method. Because pH is an important physical property of aqueous solutions, various methods have been used for pH measurements, such as glass membrane electrodes [35,36], ion-sensitive field effect transistor (ISFET) [37,38], pH indicators, and fluorescent probes [39,40]. The electrochemical method of pH measurement in which measurement of an electric current is involved is characterized by certain advantages, such as low cost, ease of operation, and ease of electrode fabrication, and is useful for developing simple sensors. Based on these advantages, there have been studies on methods for determining pH from current measurement.

pH measurement was reported in a previous study, in which the influence of the substituents on the redox reaction of ferrocene was considered [41]. This method can only be used in the range near the acid dissociation constant (pKa) of the functional groups because the pH measurement based on this principle utilizes the pH equilibrium of the weakly acidic and weakly basic functional groups. Further, compounds with quinone structures have been used for pH measurements [42,43]. Because H^+^ ions are involved in the redox reaction of quinone, the oxidation and reduction potentials shift according to the pH based on the Nernst equation. Therefore, quinone compounds can be used in a wider pH range compared with those associated with the methods based on the aforementioned principle. However, the redox potential could shift when being influenced by various compounds and pH because the quinone compounds form covalent bonds with sugars and amino acids in vivo [44].

Therefore, we decided to use Azure A (AA), which is an analog of methylene blue (MB) (Figure 1). The peak potential associated with the redox reaction of MB shifts according to the pH changes [45,46]. In addition, another study has investigated intracellular pH measurements using an MB solution [47]. However, the pH probe must be immobilized on the electrode surface for use as a sensor. Therefore, we synthesized AA-modified poly(methacrylic acid) and prepared a layer-by-layer multilayer film with poly(allylamine hydrochloride) (PAH) on a glassy carbon electrode through electrostatic interaction. The results of cyclic voltammetry (CV) and differential pulse voltammetry (DPV) measurements using (PAH/AA-PMA)_5_-modified glassy carbon electrode demonstrated that the immobilized AA participated in a redox reaction in which peak potential shifted depending on the solution’s pH. Because H^+^ ions are involved in this electrochemical response, as per the Nernst equation, the peak potential changed in a wide pH range as expressed by the following Equation (1):(1)$$E=E^{0}+\;\frac{RT}{nF}ln\frac{\left[Ox\right]{\left[H^{+}\right]}^{m}}{\left[Red\right]}$$
where *E* is the redox potential, *E*^0^ is the standard electrode potential, *R* is the gas constant, *T* is the absolute temperature, and *F* is Faraday’s constant. The measurement method like this has several advantages, such as low cost, ease of operation, and ease of electrode fabrication. Furthermore, this method allows local measurements because the probe can be minimized.

## 2. Materials

In this study, we used poly(methacryloyl chloride) (25% solution in dioxane) available from Polysciences, Inc. (Warrington, PA, USA) and PAH and AA available from Sigma-Aldrich, Inc. (St. Louis, Mo, USA). Other reagents were of special grade and used without further purification. The AA-modified PMA was prepared as follows: first, AA (55.38 mg/0.191 mmol) and triethylamine (59 mg/0.574 mmol) were dissolved in dimethylformamide (DMF, approximately 30 mL) under stirring. Next, 400 mg (corresponding to 0.957 mmol of monomer) of poly(methacryloyl chloride) (25% solution in dioxane) was diluted by gradually adding it to DMF (~10 mL) under stirring overnight. Then, adding the same volume of water as that of the solution, AA-PMA was prepared (Figure 2). The obtained polymer solution was dialyzed using a cellulose dialysis membrane with a molecular weight of 1000 as a cut-off. A part of the solution was evaporated to dry, and then the concentration of the polymer was calculated. Furthermore, the modification rate was determined by elemental analysis. Because of elemental analysis, the percentages of H, C, and N were determined as 7.42%, 50.87%, and 4.20%, respectively (C/N: 12.11). Thus, the modification rate of the resultant Azure-A-modified poly(methacrylic acid) (AA-PMA) was determined as 14.09 mol% to the methacrylic acid monomer.

## 3. Apparatus

CV and DPV were conducted using an electrochemical analyzer (ALS model 660B; BAS, Tokyo, Japan) in a conventional three-electrode cell comprising a (PAH/AA-PMA)_5_-modified electrode as the working electrode, a platinum wire as the counter-electrode, and an Ag/AgCl (3 mol L^−1^ KCl) as the reference electrode, and then the third CV cycle was recorded. DPV was then conducted at a step potential of 4 mV, a pulse amplitude of 50 mV, a pulse width of 60 ms, a pulse period of 200 ms, a sample period of 20 ms, and a voltage range from −0.6 to 0.4 V. A quartz crystal microbalance (QCA 917 system, Seiko EG & G, Tokyo, Japan) was used for gravimetric analysis of the generated multilayer films. A 9-MHz AT-cut quartz resonator coated with a thin Pt layer (surface area = 0.2 cm^2^) was used as the probe. Moreover, all measurements were performed multiple times, and average values were plotted.

## 4. Preparation of (PAH/AA-PMA)_5_-Multilayer Films

PAH solution (1.5 mL, 0.1 mg/mL, *M*_w_ 150,000) was added to a cell with a quartz resonator and left for 15 min. Then, it was washed for 5 min by adding buffer solution (1.5 mL). AA-PMA solution (1.5 mL, 0.1 mg/mL) was then added to the cell and left for 15 min. It was then washed for 5 min by adding a buffer solution (1.5 mL). Repeating the procedure, a layer-by-layer deposited film was prepared on an Au thin film. The buffer solution used for preparing layer-by-layer deposited film was 4-(2-hydroxyethyl)-1-piperazineethanesulfonic acid (HEPES, 10 mM, pH = 7.4). A glassy carbon electrode and a quartz resonator were used as the substrate. The (PAH/AA-PMA)_5_ multilayer film-modified electrode was then prepared by coating the glassy carbon electrode with the multilayer film using the same procedure.

## 5. Results and Discussion

### 5.1. Preparation of (PAH/AA-PMA)_5_-Multilayer Films

It has been known that the peak potential of a redox reaction of methylene blue (MB) changes depending on pH changes [45,46]. There has been a report of an intracellular pH measurement using a methylene blue (MB) solution [47]. However, in this pH measurement method, it is necessary to add a redox probe. Thus, in a simple measurement, it is easier to handle if the redox probe is immobilized on the electrode’s surface. Therefore, this time, we attempted to modify the electrode surface with MB by the layer-by-layer deposition method. In the experiment, PAH, which is a cationic polymer, was selected as a film material. We expected the formation of a layer-by-layer deposited film via electrostatic attraction by modifying an anionic polymer with the redox probe. Because MB has no functional groups that can bond to the polymer, AA-PMA was synthesized by adding a pendant group to poly(methacrylic acid) via covalent bonding using AA, which has a structure of MB with one of the two dimethyl amino groups being replaced with an amino group.

Figure 3 shows changes in the resonance frequency (ΔF) at each layer deposition step of the PAH/AA-PMA multilayer deposition (Note: the lines between the experimental points are only guidelines). As shown in the figure, ΔF decreased with each layer deposition step; it then became −112.3 Hz when the fifth bilayer was deposited. The amount of AA-PMA adsorption increased with the number of layers. This effect is common during the preparation of layer-by-layer multilayer films. Accordingly, we were able to confirm deposition of the cumulatively layered films. For the crystal resonator used herein, the ΔF change at 1 Hz indicates 1.1 ng of adsorption on the quartz resonator. Therefore, the amount of polyelectrolyte adsorbed after five layers of deposition was reported to be 617 ng/cm^2^.

### 5.2. pH Measurements Using the (PAH/AA-PMA)_5_-Modified Electrode

Figure 4 shows the CV measurement results when the (PAH/AA-PMA)_5_-modified electrode was used. The oxidation peak potential was ~0 V at pH = 3 and changed to −0.25 V at pH = 9. The reduction peak potential was −0.07 V at pH = 3 and changed to −0.28 V at pH = 7. The reduction peak potential was not observed at pH = 8 or pH = 9. Similar to the MB solution, the peak potential changed depending on the pH. Therefore, it was confirmed that CV measurement can be conducted even when AA was immobilized on the film via amide bonds.

### 5.3. pH Measurements Using the (PAH/AA-PMA)_5_-Modified Electrode

Similarly, DPV measurements were performed using the (PAH/AA-PMA)_5_-modified electrode (Figure 5). The peak potential was −0.03 V at pH = 3 and changed to −0.36 V at pH = 9. Furthermore, the peak potentials at pH = 8 and 9, which were not observed by CV, were confirmed by DPV. This can be attributed to the fact that the potential is applied at a constant speed in the CV measurements, whereas in the DPV measurements a pulse potential is applied while the initial potential is increased at a constant speed; furthermore, the current difference between immediately before and after the application of the potential is measured. Therefore, the effect of the charging current in the DPV measurements is reduced, and DPV allows highly sensitive measurement even when very small amount of the redox probe is used. To compare voltammograms with CV, the peak potentials in DPV are obviously sharper than those in CV, which suggests that DPV measurement is easier. Figure 5b shows the plot of the peak potentials versus pH in DPV. From this figure, it can be understood that, similar to CV, DPV can be used for pH sensing. In the slope of the line of Figure 5b, 51 mV of shift was observed per one unit of change in the pH range of 3–9 (R = 0.995). Therefore, it can be understood that DPV measurement is more suitable when using (PAH/AA-PMA)_5_-modified electrodes. In the following experiments, we studied pH sensing using DPV measurements.

### 5.4. Repeated Measurements

Next, the durability of the (PAH/AA-PMA)_5_-modified electrode as a pH sensor was evaluated. A microfabricated electrochemical sensor with good repeatability can be used in various applications, such as real-time monitoring and imaging. Using the prepared (PAH/AA-PMA)_5_-modified electrode, the DPV measurements were repeatedly conducted in phosphate buffer solution (100 mM) at pH = 3, 7.4, and 9. Consequently, the peak potential hardly changed after 50 measurements within the entire pH range, which indicates that it can be repeatedly used (Figure 6). Because AA in the film gradually became electrochemically inactive, the pAA current decreased with each repeated measurement. Thus, almost no oxidation currents were observed after 50 or more measurements. Consequently, the detection of the peak potential became difficult. Because pH is calculated from the oxidation potential of the (PAH/AA-PMA)_5_-modified electrode, this electrode was durable against repeated measurements.

### 5.5. Influence of Additives

During electrochemical measurements, the structure of the redox probe reacts with another compound and is subject to structural change. Alternatively, redox potential may change because of interactions, such as hydrogen bonding. Therefore, for pH measurements of biological samples containing sugars, salts, amino acids, and proteins, the results must not be influenced by these interferences. Generally, biological samples may contain various types of materials. DPV was conducted in a phosphate buffer solution (100 mM) containing glucose, glutamate, and the peak potentials were plotted against the pH (Figure 7). The influence of the addition of glucose or glutamate on DPV was identified when compared with samples without additives as a shift of approximately 10 mV or less. This can be attributed to a slight influence of the interaction between peak and each compound on the oxidation potential. However, the influence of additives is considered to be small because the pAA shifted by approximately 60 mV when the pH was changed by 1 unit. The redox reaction bearing quinone’s structure is used for similar pH measurements [42,43]. Although the oxidation potential of catechol in pH 7.4 phosphate buffer was about +0.2 V, which with the addition of 10 mM glucose and glutamic acid was shifted by approximately 0.04 V and 0.05 V, respectively (Figure 8). The quinone structure [48,49,50] produced by electrolytic oxidation of catechols is highly reactivity and forms covalent bonds with sugars and amino acids in vivo [44]. Therefore, the redox potential shifts significantly from the original catechol. From the above results, it was shown that (PAH/AA-PMA)_5_-modified electrode is not easily affected by interfering substances.

## 6. Conclusions

In this study, we successfully prepared PAH/AA-PMA-modified electrodes and determined pH in solutions from oxidation peaks in DPV measurements. The electrodes were usable for 50 repeated measurements, and the results were hardly affected by interfering materials. The peak potential shift of Azure A shows a wide range of pH response according to the Nernst equation, whereas the pH measurement range of PAH/AA-PMA-modified electrodes was pH 3–9. Under strongly acidic and strongly basic conditions, the PAH/AA-PMA multilayer film decomposed due to the loss of charge of PMA and PAH. However, compared with the glass membrane and ISFET electrodes, the electrode can be easily prepared and enables simple and rapid pH measurements.

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
