# Peer review of "Voltammetric pH Measurements Using Azure A-Containing Layer-by-Layer Film Immobilized Electrodes"

_polymers, 2020, doi:10.3390/polym12102328_

Round 1

Reviewer 1 Report

The manuscript by Watanabe et al investigated the pH sensing capabilities of Azure A-modified 18 poly(methacrylic acid) (AA-PMA) modified glassy carbon electrodes with assistance of typical electrochemical setup. The layer by layer technique was used to modify the GC electrode. The CV and DPV measurements shed light on this novel material-based pH sensor. The electrodes were tested for 50 usable times, still it shows the stable sensing responses. The manuscript is prepared well and presented well. Also, it showed good stability among various interferences. I have few comments before acceptance.

  1. The layer by layer technique of the materials can be explained through schematic.
  2. How the following DPV parameters were set? (4 mV, a pulse amplitude of 50 mV, a pulse width of 60 ms, a pulse period of 200 ms, a sample period of 20 ms, and a voltage range from −0.6 to 0.4 V

Author Response

Thank you for your kind suggestion of revision of our manuscript (polymers-951646). We have revised the manuscript according to reviewer’s comments. All revisions made are marked yellow background in the revised manuscript. Please see the attachment.

Reviewer 2 Report

In this paper, the authors prepared and characterized layer-by-layer films to be used as pH sensor. The obtained results are quite interesting and electrochemical response was not affected by the interfering substance considered. The manuscript is clear and can I recommend this for publication after the following revision:

- The characteristics of pH sensor as sensitivity and stability should be presented in the abstract.

- In the introduction, the authors refer the different possibilities to prepare layer-by-layers taking into account the different interactions between the molecules. In fact, wrote correctly “ Formation of a thin film can be both driven by electrostatic interaction and by other types of interactions such as biological affinity, 34 e.g. avidin–biotin bonds [2–4], sugar–lectin bonds [5]; hydrogen bonds [6, 7]; diol–phenylboronic acid bonds [8, 9]; and guest–host interactions [10].”. However, the authors forgot the pioneer´s works about this thematic, namely, the formation of layer-by-layer films due to hydrogen bonds and other low energy physical bonds published in:

            -Macromolecules, 30, 6095-6101, 1997, https://doi.org/10.1021/ma970228f

-Physica Status Solidi A- Applications and  Materials Science, 173, 41-50, 1999¸ https://doi.org/10.1002/(SICI)1521-396X(199905)173:1<41::AID-PSSA41>3.0.CO;2-Q

- Langmuir, 16(6), 2000, 2839-2844, https://doi.org/10.1021/la990945y

- In page 1, lines 39 to 40, the old references were also avoided, citing only some examples. The authors can cite some reviews about layer-by-layer films and applications to show to the readers the importance of these films.

-relatively to the presented applications of these films, I think that it should also interesting to show the interest of these films to  control of surface roughness for specific applications see for example, Microsc. Microanal. 19, 867–875, 2013, https://doi.org/10.1017/S1431927613001621 and Journal of Physical Chemistry B, 119(27) 8544-8552, 2015, 10.1021/acs.jpcb.5b02384.       

-In the caption of figure 3, it should be indicated that the lines between the experimental points are only guidelines.

-Some words should be given about the fact of the as the number of layers increases the adsorbed amount of AA-PMA also increases. This effect is common during the buildup of layer-by-layer films.

-A discussion about the effect of pH in the degree of ionization of the polyelectrolytes and in the stability of the layer-by-layer films should be included in the manuscript.

- A comparison of the sensor characteristics (e.g. sensitivity, stability,… ) with other pH sensors present is literature must be performed.

Author Response

(The authors gave the same response as above.)

Round 2

Reviewer 2 Report

The authors have improved the manuscript in accordance with the suggestions. So, now I can recommend it for publication.